# OpenReview forum: "QubitCache: Quantum-Inspired Probabilistic Attention Preservation for KV-Cache Compression"
_ICLR.cc/2026/Conference — ICLR 2026 Conference Withdrawn Submission_

### Official Review · Reviewer_8rQZ · 2025-11-01

**Soundness:** 2
**Presentation:** 1
**Contribution:** 2
**Rating:** 4
**Confidence:** 2

**Summary:**

The paper introduces a novel method that, instead of dropping tokens like prior KV cache eviction methods, encodes the attention patterns of the dropped tokens through quantum state encoding.

**Strengths:**

- The method is evaluated across a wide range of models, including large-scale models up to 70B parameters.
- Strong novelty. To my knowledge, this is the first approach that applies quantum-inspired compression to KV cache.

**Weaknesses:**

**[W1]** The paper is difficult to follow for readers without prior knowledge of quantum state encoding. Please provide a more detailed background on quantum state encoding. What is the RY rotation?

**[W2]** The paper includes some imprecise statements:
- L13: “LLM inference suffers from quadratic KV cache memory growth” --> Quadratic with respect to what? The memory growth is linear in sequence length.
- L116: Similarly, why is the memory proportional to $N^2$?

**Questions:**

**[Q1]** How were the configurations of the baseline KV cache compression methods chosen? Since their compression rate and performance heavily depend on such configurations, justifying these choices would help demonstrate that the empirical comparison is fair.

**[Q2]** How does the prefill and decoding latency of this method compare with other compression methods? Additionally, providing a throughput analysis would give a more comprehensive view.

**[Q3]** How would the method perform on reasoning models for long-generation tasks such as AIME or LiveCodeBench? Demonstrating this would highlight the method’s robustness under long-generation scenarios, which are not captured by the short and long-context tasks.

---

### Official Review · Reviewer_kvp5 · 2025-11-01

**Soundness:** 3
**Presentation:** 3
**Contribution:** 3
**Rating:** 4
**Confidence:** 3

**Summary:**

The paper proposes QubitCache, a hybrid quantum-inspired KV-cache compression method that encodes attention patterns (rather than raw token vectors) using amplitude-style encodings and probabilistic reconstruction, while keeping a small set of critical tokens in classical memory. The authors argue this preserves relational structure across long contexts and report a 7× reduction in KV memory with 92–97% retention of baseline performance across several models and long-context benchmarks; they support this with ablations, reconstruction metrics, and qualitative generation comparisons using a 9-qubit segment encoding simulated with Qiskit on GPUs.

**Strengths:**

* The paper motivates encoding attention patterns instead of performing binary token eviction, and ties this idea to graph-theoretic and spectral arguments that are well described and supported by references and analysis.

* Experiments cover multiple model families and sizes, several long-context benchmarks, ablation studies, reconstruction metrics, and qualitative samples, which together give a reasonably complete picture of where the method helps and where it does not. Reported gains on multi-hop reasoning and the memory numbers are impressive if reproducible.

**Weaknesses:**

* Hardware and complexity claims feel optimistic relative to current quantum realities. The paper’s argument that 9-qubit, depth-15 circuits are directly compatible with NISQ devices and that noise can be leveraged as regularization requires stronger evidence; the experiments are classical simulations and do not demonstrate real-device results or latency/throughput comparisons that would matter in production. The extrapolation to hardware acceleration therefore remains speculative.

* Important implementation details for baselines, shot counts, measurement variance, and statistical significance of reported differences are missing or relegated to the appendix. Given the fairly large claims (7× memory, 15–25% higher F1 on multi-hop), the paper should present variance estimates, run multiple random seeds or splits, and make code/configs available to ensure the results are robust and reproducible.

* The approach trades memory for additional computation and potential latency, but the runtime and energy costs are not quantified. The paper gives asymptotic and GPU memory figures but does not provide wall-clock end-to-end inference latency, throughput under batching, or the cost of repeated re-encoding and measurement in streaming generation. These metrics are crucial for deployment decisions and could change whether the method is practically preferable to strong classical baselines.

**Questions:**

Please see weaknesses

---

### Official Review · Reviewer_wZek · 2025-11-01

**Soundness:** 2
**Presentation:** 3
**Contribution:** 2
**Rating:** 2
**Confidence:** 3

**Summary:**

The paper proposes QubitCache, a method that retains 15% of tokens, comprising 10% of the most recent tokens, 5% of tokens with high attention scores, and 4 anchor tokens. For the remaining tokens, the model reconstructs the pruned value vectors by interpolating between neighboring value vectors and reconstructs attention scores using quantum amplitude encoding. The authors compare QubitCache with ScissorHand, H2O, StreamingLLM, and GEAR baselines on both short- and long-context understanding tasks.

**Strengths:**

The paper introduces a novel perspective on KV cache compression by leveraging quantum-based encoding and reconstructing the value vectors through interpolation.

**Weaknesses:**

- Concerns about experiments
    - The paper is missing several state-of-the-art baselines, such as SnapKV [1] and PyramidKV [2].
    - In Table 1, the compression ratio is not consistent across baselines and results are only reported for a single compression ratio for each method. It would be more informative to include a plot showing accuracy versus compression ratio (with the x-axis representing the compression ratio and the y-axis representing accuracy).
    - Additionally, based on the memory compression ratios reported in Table 3, QubitCache retains at least 15% of the cache, implying a maximum possible compression ratio of approximately 100 / 15 = 6.67x. However, the table reports a ratio of 7.0x, which appears to be inconsistent.
- Concerns about the method
    - The proposed method reconstructs value vectors at inference time by interpolating neighboring value vectors and extracting attention patterns. However, the paper does not provide any discussion or measurement of the inference-time latency or acceleration ratio, which is essential for understanding the practical efficiency of the approach.
    - Equation (6) seems to assume that value vectors are linearly aligned, which is unlikely to hold in practice. It would be valuable to verify this assumption through statistical analysis, as it may not be generally true.
    - Furthermore, it would be interesting to explore simpler alternatives that do not rely on quantum-based amplitude encoding for reconstructing attention pattern. For instance, one could reconstruct value vectors using an averaged representation (e.g., $\bar{V} = \sum_{j \in I_C} \tilde{V}_j$) with a weighting factor ($1 - \lambda$), or apply this in a fine-grained, group-wise manner (e.g., over every (g) tokens). Such comparisons would help clarify whether the quantum-inspired encoding provides a meaningful advantage over simpler interpolation strategies.
- Minor
    - The first sentence of the abstract is incorrect. The KV cache memory grows linearly with the sequence length, whereas the total FLOPs computed phase grow quadratically.

[1] Li, Yuhong, et al. "Snapkv: Llm knows what you are looking for before generation." NeurIPS 2024.

[2] Cai, Zefan, et al. "Pyramidkv: Dynamic kv cache compression based on pyramidal information funneling.", ACL 2024 Findings.

**Questions:**

See the above weakness section.

**Details Of Ethics Concerns:**

I do not have ethics concern regarding this paper.

---

### Official Review · Reviewer_CkeY · 2025-11-04

**Soundness:** 3
**Presentation:** 2
**Contribution:** 3
**Rating:** 6
**Confidence:** 2

**Summary:**

The authors propose a KV cache compression scheme inspired by quantum encoding. A handful of tokens (sink, recent, attention-critical within middle context) are preserved while the remainder are retained via amplitude encoding. The attention patterns of non-preserved tokens are probabilistically restored. This scheme ultimately enables logarithmic compression based on the number of circuit qubits required for sufficient accuracy. This approach enables 7x memory reduction while enabling high performance on a variety of long-context and short-context tasks against baselines such as H2O, Scissorhands, and GEAR.

**Strengths:**

- A fresh, quantum-inspired approach to cache compression. Offloading approaches such as MagicPIG probabilistically restore attention distributions as well, but require full-cache maintenance on the CPU, unlike this work.

- The method is empirically verified to perform well on several challenging LongBench tasks and other short-context tasks against popular strategies such as H2O and Scissorhands.

- The approach seems to generalize across model families.

**Weaknesses:**

- Figure 2 will not be understood by the lay ML reader. More effort needs to be explaining these circuits a general ML audience.

- The theoretical analysis is oversold -- the abstract and introduction implies there is some rigorous guarantee or theorem, but there is no such exposition in the primary manuscript.

- While the viability of logarithmic compression is empirically observed, it would useful to see when this approach breaks (i.e., at what number of qubits for encoding versus segment length does this approach break).

- How were the LongBench tasks chosen? I'm curious as to how this approach performs on a challenging task such as LCC (long code completion?).

- It is relatively well-known that existing long context benchmarks such as RULER and LongBench are easy to solve since critical tokens are sparsely distributed (i.e., lots of gibberish/noise within the context) , as opposed to benchmarks such as GSM-Infinite which contain less noisy context.

**Questions:**

See weaknesses.

---

### Note · Authors · 2025-11-14

I have read and agree with the venue's withdrawal policy on behalf of myself and my co-authors.